# *Bacillus velezensis*: A Beneficial Biocontrol Agent or Facultative Phytopathogen for Sustainable Agriculture

**Muhammad Fazle Rabbee** [1,†]**, Buyng-Su Hwang** [2,†] and **Kwang-Hyun Baek** [1,*]

1    Department of Biotechnology, Yeungnam University, 280 Daehak-Ro, Gyeongsan 38541, Republic of Korea
2    Nakdonggang National Institute of Biological Resources, 137 Donam 2-gil, Sangju-si 37242, Republic of Korea
*    Correspondence: khbaek@ynu.ac.kr; Tel.: +82-53-810-3029
†    These authors contributed equally to this work.

**Abstract:** Microbial biocontrol agents are efficient and environment-friendly in eradicating plant pathogenic bacteria. In recent years, *Bacillus velezensis* has gained popularity as a potential biocontrol agent in many countries. Several *B. velezensis*-based biocontrol products, previously identified as *B. amyloliquefaciens* or *B. amyloliquefaciens* subsp. *plantarum*, have received commercial approval, particularly in China and Europe. In this study, we compiled recent research findings on *B. velezensis* related to the production of antimicrobials, volatile organic compounds, induction of disease resistance, and the effect of this bacterium on plant growth promotion and yield. However, some recent research indicates that this important resource is also linked to several diseases in crops, including peach, onion, and potato, and the negative aspects of this bacterium in terms of its virulence traits to infect crops have not been summarized before. In this review, we compile the recent reports of this bacterium in term of its beneficial properties in agriculture. In addition, we also discuss several reports about its harmful effect on several crops as well. Therefore, due to the inherent pathogenicity of this bacterium to several crops, care must be taken when using it in a novel crop cultivation technique.

**Keywords:** *Bacillus velezensis*; sustainable agriculture; benefits; deleterious effect





## 1. Introduction

The world is facing a strong need to develop ecofriendly and sustainable methods to improve agricultural productivity. Microbial infections of plants pose a primary danger to global food production and healthy ecosystems. Plant pathogenic microorganisms are estimated to cause a ~25% loss of the global crop yield each year [1]. In addition, over the last 50 years, the human population has doubled, and by 2050, it is expected to reach over nine billion [2]. To fulfill the food demand of this growing population, a considerable enhancement of agricultural production is required [2]. Changing environmental conditions with the reduction of agricultural farmlands also pose a negative impact on meeting the food demand of the increasing population [3]. Moreover, modern agriculture relies heavily on the intensive use of agrochemicals, such as fertilizers and pesticides, which have detrimental effects on both the environment and human health [4]. To replace chemical agents in agriculture, many studies have attempted to identify novel bacterial strains that can be used as biocontrol and/or biofertilizer alternatives [5]. In 2020, the market of global biocontrol agent reached to USD 4.0 billion, and the demand is projected to reach USD 10.6 billion by 2027 [6].

Microbes are detected in almost every ecosystem; thus, all plant and animal species live on the planet of microbes [7]. Based on how microbes interact with their plant hosts, plant-associated microbes can be categorized into three groups: helpful, harmful, and neutral. These microbes continually interact with the host plants and among themselves to form the plant microbiome [8]. Plant microbiomes linked to increased plant health and productivity are emerging as alternatives to agrochemical-based approaches. For example,

in recent years, endophytic microbes are gaining attention of researchers and scientists due to their ability to produce metabolites that regulate host plant physiology or having pharmacological significance [9]. Endophytes are microbes (i.e., bacteria or fungi) that reside in plants asymptomatically. Endophytes may be responsible for host resistance to pathogens through the synthesis of antimicrobial compounds [10]. Endophytic microbes have been demonstrated to (i) obtain nutrients from the soil and transfer them to plants via the rhizophagy cycle and other nutrient-transfer symbioses, (ii) boost plant growth and development, (iii) reduce host oxidative stress, (iv) protect plants from disease, (v) discourage herbivore feeding, and (vi) suppress the growth of rival plant species [11]. Some endophytic plant-growth-promoting bacteria can induce multidrug resistance and virulence genes through horizontal gene transfer mechanisms to other bacterial communities. Microbes enter the host plant to perform beneficial functions in it as an endophyte; however, the underlying mechanisms of plant defense by endophytes to achieve harmonious commensalism remains unclear [7]. *B. subtilis* BSn5 reduces plant defensive responses through the production of the antibiotic subtilomycin, which masks self-produced flagellin, enabling better colonization of BSn5 in *Arabidopsis thaliana* [7].

Plant-beneficial microorganisms increase agricultural productivity and serve as alternatives to artificial pesticides and fertilizers by protecting plants against diseases. Research is ongoing to identify new microbes from a variety of sources that can be used as biocontrol agents. *Bacillus* and *Pseudomonas* are the most widely used biological alternatives to chemical agents. *Bacillus*-based agro-alternatives are gaining attention owing to their spore-forming ability that can resist harsh environmental conditions for an extended period [12]. In addition, *Bacillus* species reportedly have biocontrol potential in the greenhouse, field, and post-harvest stages of fruit because of their contribution to plant protection through several mechanisms, including antibiosis, reduced pathogen colonization in the root via competition, and induction of systemic resistance (ISR) in the host plant [13].

In this study, among many known *Bacillus* species, the roles of recently characterized *B. velezensis* are investigated considering its extensive potential usage in the production of useful biomaterials as well as its high biocontrol capacity. In addition, we summarize the recent reports about the possibility of disease incidence caused by several *B. velezensis* strains in potato, onion, and peach fruits. To our knowledge, this is the first report highlighting the negative aspects of employing *B. velezensis* in agriculture.

## 2. Role of *B. velezensis* in Controlling Plant Pathogens

Technological developments have prompted a revival of the search for natural antibiotics from bacterial sources, leading to the discovery of various novel antibiotics with distinctive scaffolds and novel mechanisms of activity to serve as the foundation for novel antibiotic classes targeting pathogenic microorganisms [14]. Several *Bacillus*-based biocontrol agents, including *B. velezensis*, *B. subtilis*, *B. amyloliquefaciens*, and *B. thuringiensis*, remarkably control plant pathogens by secreting a variety of antimicrobial compounds, including lipopeptides (LPs), polyketides (PKs), antibiotics, and enzymes, through colonization [15,16]. Among these biocontrol agents, *B. velezensis* can produce diverse secondary metabolites by producing nonribosomal peptide synthetases (NRPS) and polyketide synthases (PKS) [17]. The effects of NRPS- and PKS-derived antimicrobial compounds from *B. velezensis* on plant pathogens are summarized in Tables 1 and 2.

**Table 1.** Comparison of gene clusters encoding antimicrobial compounds in several commercial *B. velezensis*-based biocontrol agents. The gene clusters in *B. velezensis* QST71 (Serenade®) are compared with the genomes of *B. velezensis* GB03 (Kodiak TM) and *B. velezensis* 83 (Fungifree AB™).

| | Gene Clusters of *B. velezensis* QST713 (NCBI Accession Number: CP025079.1) | | | | Bioactive Compounds Presence (+)/Absence (−) | |
|---|---|---|---|---|---|---|
| Region | Bioactive Compounds | Type | Size (nt) | Similarity | GB03 (CP049904.1) | 83 (CP034203.1) |
| 1 | Rhizocticin A | NRPS/PKS | 78,608 | 22% | - | - |
| 2 | Surfactin | NRPS | 66,095 | 86% | + | + |
| 3 | Butirosin A/Butirosin B | PKS | 42,244 | 7% | + | + |
| 4 | Unknown | Terpene | 18,104 | - | - | - |
| 5 | Macrolactin H | PKS | 88,814 | 100% | + | + |
| 6 | Bacillaene | PKS/NRPS | 100,882 | 100% | + | + |
| 7 | Fengycin | NRPS | 137,406 | 100% | + | + |
| 8 | Unknown | Terpene | 22,883 | - | - | - |
| 9 | Unknown | PKS | 42,100 | - | - | - |
| 10 | Difficidin | PKS | 94,798 | 100% | + | + |
| 11 | Bacillibactin | NRPS | 51,501 | 100% | + | + |
| 12 | Subtilin | Lanthipeptide class-I | 27,785 | 100% | Lanthipeptide class II: Mersacidin | Lanthipeptide class III: Locillomycin |
| 13 | Unknown | NRPS | 69,429 | - | - | - |
| 14 | Bacilysin | NRPS/PKS | 42,418 | 100% | + | + |

### 2.1. Antimicrobial Substances Synthesized by B. velezensis to Control Plant Pathogens

*B. velezensis* is an endospore-forming and free-living soil bacterium first described by Ruiz-Garcia in 2005, and was previously classified as another *Bacillus* species before being reconfirmed as *B. velezensis* [18]. *B. velezensis* produces various secondary metabolites that act against different phytopathogenic microbes. For example, whole-genome analysis revealed that *B. velezensis* HNA3 had 12 gene clusters responsible for the synthesis of 14 secondary metabolites. Among the metabolites, five nonribosomal LPs are fengycin, bacillomycin D, surfactin, mycosubtilin, and bacillibactin; three PKs are macrolactin, difficidin, and bacillaene; two antimicrobial peptides are bacilysin and amylocyclicin; and other secondary metabolites include plipastatin, iturin, paenibactin, and paenilarvins [17]. Another genome analysis of *B. velezensis* QST713 (previously reported as *B. subtilis* QST713) revealed that at least 12% of the genome of this bacterium is involved in the biosynthesis, regulation, and transport of LP and PK molecules (Figure 1). In addition to the antimicrobial compounds found in *B. velezensis* FZB42 or *B. velezensis* SQR9, this bacterium contains two additional gene clusters that encode rhizocticin- and subtilin-like compounds (ericin) [19].

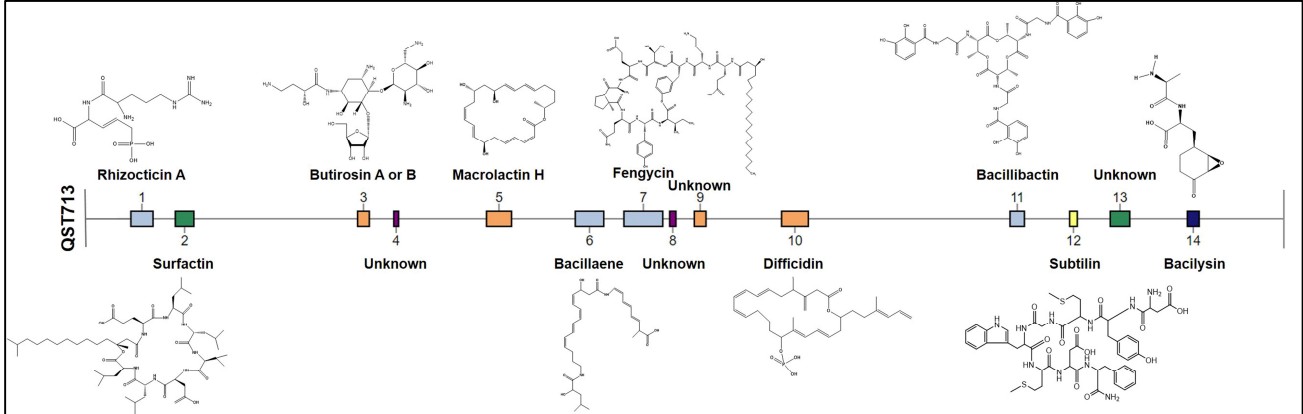

**Figure 1.** Antimicrobial gene clusters identified in the genome of the *B. velezensis* QST713 strain.

Antimicrobial compounds produced by *B. velezensis* have a wide range of applications in controlling plant pathogenic microbes. For instance, *B. velezensis* exerts antagonistic effects against *Ralstonia solanacearum* and *Fusarium oxysporum* through lipopeptides, including surfactin, iturin, and fengycin [20]. *Bacillus* spp. wsm-1 isolated from the deep sea exhibited

strong antifungal action against several pathogenic fungal strains (e.g., *Magnaporthe grisea*, *Fusarium oxysporum*, *Colletotrichum fioriniae*, and *Alternaria alternata*) due to the presence of novel lipopeptide molecule $C_{14}$ iturin W and $C_{15}$ iturin W [21]. *B. velezensis* QST713 is used industrially as a biocontrol agent in France for the protection of *Agaricus bisporus* (an edible mushroom) in a compost micromodel to control *Trichoderma aggressivum* f. *europaeum*, which causes green mold disease. Transcriptomic analysis revealed the upregulation of several genes, including surfactin (*srf* AA) and fengycin (*fen* A), in the presence of fungal pathogens in compost [22]. The *B. methylotrophicus* NKG-1 strain inhibited the growth of tomato mold disease caused by *Botrytis cinerea* by 60% [23]. Fluorescent-labeled endophytic *B. velezensis* CC09, widely distributed in wheat plant tissues including the cortex and xylem vessels, as well as stems and leaves, conferred the disease control efficacy of 66.67% and 21.68% in the take-all disease caused by *Gaeumannomyces graminis* var. *tritici* and spot blotches of wheat leaves caused by *Bipolaris sorokiniana*, respectively [24]. Rhizobacterium *B. velezensis* SQR9 produces a unique antibacterial compound, bacillunoic acid, that acts against closely related *Bacillus* spp. and aids in the formation of a self-protecting shield, resulting in increased competition in the rhizosphere zone of the root [25]. We previously demonstrated the bactericidal activity of ethyl acetate extract of *B. velezensis* Bv-25 against wild-type and streptomycin-resistant strains of *Xanthomonas citri* subsp. *citri* (*Xcc*), causing citrus canker disease [26,27].

### 2.1.1. Phytopathogens Controlled by Difficidin and Bacilysin

Bacterial blight and leaf streak infection caused by *X. oryzae* pv. *oryzae* and *X. oryzae* pv. *oryzicola*, respectively, were successfully controlled by difficidin and bacilysin produced by *B. velezensis* FZB42 [23]. These compounds downregulated the genes related to pathogenicity, cell division, and cell wall formation of pathogens [28]. Bacilysin inhibits the enzymatic activity of glucosamine 6-phosphate synthase, which aids in the production of glucosamine 6-phosphate from fructose-6-phosphate and glutamine, two vital components of the bacterial cell wall peptidoglycan [29,30]. Bacilysin production in *B. velezensis* FZB42 damaged the hyphal structure of *Phytophthora sojae* (causing soybean root rot disease), leading to the loss of intracellular content [26]. However, several *B. velezensis* FZB42 mutants deficient in the production of LPs (bacillomycin D and fengycins) and PKs (difficidin, macrolactin, and bacillaene) did not exhibit antagonistic effects against *P. sojae* [31].

### 2.1.2. Phytopathogens Controlled by Bacillomycin D

*B. velezensis* LM2303 exhibited strong antifungal activity against *F. graminearum* and suppressed the disease incidence of *Fusarium* head blight, with a control efficiency of 72.3% under greenhouse conditions [32]. The *B. velezensis* SQR9 mutant, which lacks a gene cluster encoding bacillomycin D, displayed minor antagonistic action against *F. oxysporum* (responsible for vascular wilt in cucumber plants), suggesting that bacillomycin D has antifungal activity against *F. oxysporum* [33]. Additionally, bacillomycin D produced by wild-type *B. velezensis* SQR9 serves as a signaling molecule in the development of biofilms through the acquisition of iron molecules [28]. Another study showed that bacillomycin D stimulates the transcription of the iron ABC transporter (FeuABC) by binding to its transcription factor Btr (iron transport regulator). In this mechanism, SQR9 raises the level of intracellular iron and activates KinB-Spo0A-SinI-SinR-dependent biofilm matrix component production [34,35]. In biofilm formation, microbial cells aggregate to form a collective living mass embedded in a self-produced extracellular matrix [36]. Bacillomycin D produced by *B. velezensis* FZB42 exhibited strong antifungal activity against *F. graminearum*, which causes *Fusarium* head blight. Electron microscopy analysis revealed that bacillomycin D (30 μg/mL) causes exterior damage to fungal hyphae and conidia with irregular shapes, loosening of cell walls, and shriveled trunks [37]. The iturin family members, bacillomycin D and iturin A, share structural similarities with the seven amino acids in the cyclic lipopeptide molecule [38]. AI-2 synthetase translated from *luxS* in *B. velezensis* SQR9 is responsible for quorum sensing; therefore, deletion of this gene decreased the ability to build biofilms, motility, and root

colonization in the mutant strains. Adding this gene to the mutants restored the wild-type function, indicating that AI-2 positively influences root colonization of the SQR9 strain by quorum sensing [39].

### 2.1.3. Phytopathogens Controlled by Fengycins

The fengycin family comprises β-hydroxy fatty acids with 16–19 carbon atoms long side chains [33]. Fengycin has two isomers, fengycin A and fengycin B. Fengycin A contains one D-alanine, one L-isoleucine, one L-proline, one D-allo-threonine, two L-glutamatic acids, one L-glutamine, one D-tyrosine, one L-tyrosine, and one D-ornithine, whereas fengycin B has one D-valine in place of one D-alanine [38]. Fengycins are a class of antifungal LPs compounds commercially available as the primary component of the fungicide Serenade® obtained from *B. velezensis* QST713 [40]. Fengycin produced by *B. velezensis* suppressed mycelial proliferation of *F. solani* with a half-maximal inhibitory concentration (IC$_{50}$) of 5.58 µg/mL, superior to that of two commercial fungicides *viz.* thiram (IC$_{50}$ 41.24 µg/mL) and hymexazol (IC$_{50}$ 343.31 µg/mL) [41]. Fengycins presumably change the shape and permeability of the fungal cell membrane by interacting with membrane sterols and phospholipid molecules [42]. Fengycin isolated from *B. velezensis* Bs006 at a concentration of 50 µM in potato dextrose medium significantly inhibited the development of the hyphae of *F. oxysporum* f. sp. *physali*, which causes *Fusarium* wilt disease in golden berries [43].

**Table 2.** Antimicrobial compounds from *B. velezensis* acting against phytopathogens.

| *B. velezensis* Strains | Antimicrobial Compounds | Target Pathogens (Plant Diseases) | References |
|---|---|---|---|
| *B. velezensis* FZB42 | Bacillomycin D | *F. graminearum* (*Fusarium* head blight) | [37] |
| *B. velezensis* FZB42 | Difficidin and bacilysin | *X. oryzae* pv. *oryzae* (bacterial blight of rice); *X. oryzae* pv. *oryzicola* (bacterial leaf streak) | [28] |
| *B. velezensis* FZB42 | Bacilysin | *P. sojae* (soybean root rot disease) | [31] |
| *B. velezensis* Y6 and F7 | Surfactin, Iturin and Fengycin | *R. solanacearum* (tomato wilt); *F. oxysporum* (banana wilt) | [20] |
| *B. velezensis* SQR9 | Bacillomycin D | *F. oxysporum* (cucumber vascular wilt) | [33] |
| *B. velezensis* MEP$_2$18 | Fengycins | *X. axonopodis* pv. *vesicatoria* (bacterial spot disease) | [44] |
| *B. velezensis* G341 | Bacillomycin L and Fengycin A | *Alternaria panax*, *F. oxysporum*, *B. cineria*, *Phytophthroa capsici* | [45] |
| *B. velezensis* WRB-ZX-001 and WRB-ZX-002 | Iturin A | *Aspergillus fumigatus* | [46] |
| *B. velezensis* CC09 | Iturin A | *G. graminis* var. *tritici* (take-all disease) and *B. sorokiniana* (spot blotch of wheat) | [24] |
| *B. velezensis* FJAT-46737 | Fengycins | *R. solanacearum* | [47] |
| *B. velezensis* | Fengycins | *F. solani* (basal stem rot in passion fruit) | [41] |
| *B. velezensis* NJN-6 | Bacillomycin D | *F. oxysporum* | [48] |
| *B. velezensis* NJN-6 | Macrolactin | *R. solanacearum* | [48] |

Note: Several *B. amyloliquefaciens* strains, such as *B. amyloliquefaciens* SQR9, *B. amyloliquefaciens* MEP$_2$18, and *B. amyloliquefaciens* FZB42, were reclassified and reported as *B. velezensis*.

### 2.1.4. Bacteriocins Produced by *B. velezensis*

Bacteriocins in several bacterial species are ribosomally synthesized proteinaceous substances with antimicrobial effects on other bacteria or occasionally against closely related species of producer strains [49]. Bacteriocins generally act on the bacterial cell wall by inducing pore formation or inhibiting cell wall biosynthesis in target cells [50]. Bacteriocins have four classes: class I, including peptides that are ribosomally produced and undergo extensive post-translational modifications; class II, including small heat-stable and unmodified peptides; class III, including large antimicrobial peptides (now known as bacteriolysins); and class IV, including complex bacteriocins containing carbohydrate or lipid moieties [49]. Recently, lactococcin Lcn972, a novel class II bacteriocin, was discovered from *B. velezensis* HN-Q-8 and expressed in *E. coli* to obtain the compound. This bacteriocin

exerted a significant inhibitory effect (minimum inhibitory concentration 10.58 μg/mL) against *Streptomyces scabies*, which causes common potato scab disease. The stability test showed that Lcn972 is highly stable against UV radiation, high temperature (4–121 °C), and long-term storage at room temperature for 16 days [51]. *B. velezensis* FZB42 produces plantazolicin, a ribosomally synthesized complex molecule that exhibits antibacterial effects against bacterial and fungal pathogens. A cluster of 12 genes in *B. velezensis* FZB42 is essential for the production, modification, export, and self-immunity of this natural compound [52]. Amylocyclicin is another ribosomally synthesized bacteriocin by *B. velezensis* FZB42, and a cluster of six genes is responsible for its production, modification, export, and self-immunity [53].

## 3. Bioactive Enzymes Produced by *B. velezensis*

*Bacillus* species are known for the production of numerous types of extracellular enzymes involved in self-defense, metabolic support, and maintaining a normal physiological state [48]. However, this genus is the most important source of proteases and proteolytic enzymes resistant to temperature, pH, organic solvents, and oxidizing enzymes [54]. *B. velezensis* BS2, isolated from sea squirt jeotgal, produced fibrinolytic enzymes (131.15 mU/μL) at 96 h in tryptic soy broth. From *B. velezensis* BS2, the gene *aprEBS2* encoding the main fibrinolytic protein was cloned and overexpressed in *B. subtilis* WB600. The modified *B. subtilis* WB600 strain exhibited 1.5 times as much fibrinolytic activity as the wild-type *B. velezensis* BS2 [49]. The purified protein exhibited strong α-fibrinogenase and moderate β-fibrinogenase activities.

Chitinases have attracted attention lately as potential biopesticides because they can control insects, fungi, and nematodes simultaneously. Chitinases can break down a variety of pest tissues, including the peritrophic matrix and cuticle in insects, eggshells in nematodes, and cell walls of fungal phytopathogens. Chitin is a major constituent of the fungal cell wall and consists of β-1-4-linked *N*-acetyl-D-glucosamine units, the most abundant polymers in nature after cellulose [55]. Chitinase exhibits antifungal action through the disruption of fungal cell wall structures; however, the growth of chitinase-resistant fungi is unlikely to occur. Chitinase enzymes catalyze the hydrolysis of glycosidic linkages and convert chitin into water-soluble chitin oligosaccharides [56]. When the *chiA* gene-encoding chitinase from *B. velezensis* was expressed in *E. coli*, the recombinant rBvChiA protein displayed antifungal activity against *F. falciforme*, the causative agent of black pepper disease. Therefore, *B. velezensis* could be developed as a chitinolytic bacterium for efficient crop protection against pathogenic fungi and the reduction of chemical pesticides to control pests [57].

## 4. Production of Volatile Organic Compounds (VOCs) and Induction of ISR by *B. velezensis*

Microbial VOCs are organic compounds that play a key role as signals in intra- and inter-kingdomic interactions over distances of >20 cm. These small compounds (<300 Da), containing up to two functional groups, can easily diffuse in air and water [58]. VOCs produced by bacteria include alcohols, carbonyl compounds, hydrocarbons, aromatic compounds, and sulfur- and nitrogen-containing compounds, all of which exhibit a broad range of structural diversity [58]. *B. velezensis* strains can produce several VOCs, as shown in Figure 2. The VOCs produced by *B. velezensis* NGJN-6 (NCBI accession no. CP007165.1) inhibited the growth of *F. oxysporum* f. sp. *cubense*, which causes soil-borne fungal diseases. The development of *F. oxysporum* was entirely suppressed by two VOCs, benzothiazoles phenol and 2,3,6-trimethyl phenol [59]. VOCs produced by *B. velezensis* VM11 exhibited antifungal activity against *Sclerotinia sclerotiorum* through the deposition of reactive oxygen species in mycelial cells. Transmission electron microscopy analysis revealed ultrastructural malformations in *S. sclerotiorum* through the loosening of cell walls, swelling of vacuoles, loss of cell walls, disintegration of hyphal walls, and movement of cytoplasmic materials [60]. *B. velezensis* ZSY-1 produced 29 distinct VOCs as identified using GC-MS analysis,

and these VOCs included pyrazine (2, 5-dimethyl), benzothiazole (4-chloro-3-methyl), and phenol-2,4-bis (1,1-dimethylethyl), which showed antifungal effects against *A. solani* and *B. cinerea*. These VOCs are also considered promising biocontrol agents for controlling tomato fungal diseases, such as early blight and gray mold [61].

**Figure 2.** The structure of beneficial compounds other than LP- and PK-type addressed in this article. VOCs: benzothiazoles phenol; 2,3,6-trimethyl phenol; pyrazine (2,5-dimethyl); benzothiazole (4-chloro-3-methyl); phenol-2,4-bis (1,1-dimethylethyl); ISR induction: 2,3-butanediol; growth promotion: IAA and GA3; pharmaceutical compound: lobetyolin.

Plants respond to various types of diseases through ISR, which increases the expression of defense-related genes to protect against a variety of plant pathogens. *Bacillus* species can elicit systemic resistance in plants through the secretion of different types of molecules. *B. velezensis* can protect host plants by triggering an immune response in host organs through ISR. For instance, *B. velezensis* CC09, when injected into the roots of wheat plants, acts as a "vaccine" that shields them from two leaf and root diseases. Another VOC, 2,3-butanediol, produced by *B. velezensis* can produce ISR in the host plant. The antifungal activity of iturin A may be an indirect effect of systemic resistance generated by *B. velezensis* CC09 in wheat plants [24]. Endophytic *B. velezensis* YC7010 enhanced plant defense against the brown plant hopper, one of the most serious insect pests reducing rice yield remarkably. Transcriptome analysis revealed that *B. velezensis* elicits ISR via salicylic-acid- and jasmonic-acid-dependent pathways. Further research showed that ISR was induced by the novel lipopeptide compound, bacillopeptin X [62].

*B. velezensis* GJ11 may induce potent ISR against *P. syringae* pv. *tomato* DC3000 infection in *A. thaliana* by producing acetoin (3-hydroxy-2-butanone). When the genes *bdh* (2,3–butanediol dehydrogenase) and *gdh* (glycerol dehydrogenase) were knocked out to create mutant strains of GJ11, only the GJ11Δ*bdh* strain produced high levels of acetoin. The host plants treated with GJ11Δ*bdh* triggered strong ISR against the pathogenic *P. syringae* pv. *tomato* DC3000 [63]. *B. velezensis* can trigger the genes for the accumulation of plant-defense-related compounds (such as hydrogen peroxide) and defense enzymes (such as SOD, CAT, and POD), protecting the host pepper leaves against gray mold disease caused by *B. cineria* [64]. qPCR analysis revealed that applying *B. velezensis* QST713-based biofungicide induced the expression of host plant defense-related genes in canola plants [60].

The canola plants treated with this QST713-based biofungicide exhibited increased gene expression for numerous plant-defense-related pathways, including ethylene, jasmonic acid, and phenylpropanoids, by 2.2–23 folds compared with the controls treated with only distilled water [65]. In addition, the expression of *PR* genes, such as *MdPR1* and *MdPR5*, in response to apple fruit treatment by *B. velezensis* P2-1 did not reduce the quality of apple fruits [66].

## 5. Plant Health and Growth Promotion by *B. velezensis*

*B. velezensis* T20E-257, isolated from tomato, can produce 2,3-butanediol, which aids in the promotion of plant growth and health when applied as a pure compound [58,67]. Inoculation of the crop with *B. velezensis* WRN031 promotes plant growth [68]. Fluorescence microscopy indicated that GFP-labeled *B. velezensis* WRN031 accumulated in the maturation zones of the primary and lateral roots of maize. Furthermore, the presence of two nonvolatile stereoisomers of acetylbutanediol in the soil rhizosphere greatly increases rice and maize root length [68]. The genome of *B. velezensis* HNA3 contains genes involved in the production of indole acetic acid (IAA; also referred to as auxin), responsible for plant growth promotion [17]. The application of *B. methylotrophicus* NKG-1 (actual species as *B. velezensis*) to tomato seedlings increased seedling fresh weight (by 27.4%), seedling length (by 12.5%), and root length (by 57.7%), indicating the strain's potential as a biofertilizer or biocontrol agent in the commercial sector [23]. *B. velezensis* SQR9 in cucumber roots enhanced the secretion of D-galactose (chemoattractant), a signal for the interactions between SQR9 and host plants. D-galactose is crucial for bacterial root colonization and plant-growth-promoting activities in the rhizosphere [69].

*B. velezensis* SQR9 colonizes plant roots and secretes metabolites to attract indigenous plant-beneficial bacteria such as *P. stutzeri* XL272 (Figure 3). This dual consortium forms biofilms and shares the extracellular matrix and metabolites, thus promoting plant growth and alleviating salt stress [70]. Colonization of *B. velezensis* SQR9 into the plant root is very important for their specific function, which is controlled by the two-component signal transduction system DegS/U. To improve colonization into the root, scientists recently constructed a genetically engineered xylose-inducible degQ strain *B. velezensis* SQR9XYQ [71]. Green house experiments and RT-qPCR analysis indicated that phosphorylation of DegU can be activated by xylose present in cucumber and tomato root exudates. In addition, root colonization, biofilm formation, and biocontrol efficiency were greatly improved in the engineered strains compared to wild-type strain SQR9 [71]. Tomato plants treated with *B. velezensis* 83 in greenhouse cultivation produced 254.0 tons/hectare/year (64.0% fresh quality tomato), which was 184.0 tons/hectare/year (55.0% fresh quality tomato) in control without *B. velezensis* treatment [72]. The application of rhizobacterial *B. velezensis* GB03 increased the growth of *Codonopsis pilosula*, a traditional Chinese herbal medicinal plant. In addition, the treatment of medicinal plants with *B. velezensis* GB03 doubled the concentration of the secondary metabolite lobetyolin (polyacetylenes), used for treating stomach ulcers [73], and also increased the amino acids in the roots. The application of *B. velezensis* NJN-6 to banana plants has beneficial effects, including biocontrol potential against banana *Fusarium* wilt disease through the secretion of the LPs compound iturin A. Further, plant-growth-promoting hormones, such as IAA and gibberellin A3 (GA3), were detected when the *B. velezensis* NJN-6 strain was incubated with Landy broth with L-tryptophan and in root exudates of banana plants [74].

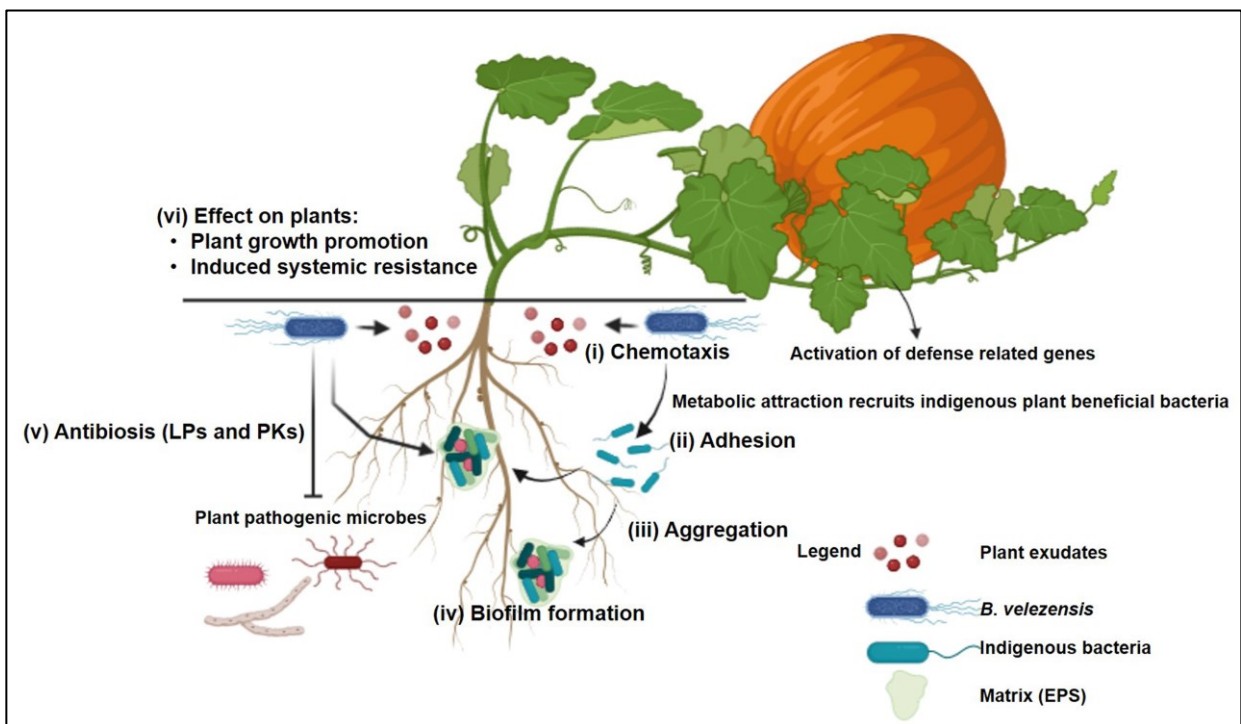

**Figure 3.** Sequential events that take place in the rhizosphere following the application of biocontrol agent *B. velezensis*. Root colonization of *B. velezensis* through (i) chemotaxis, (ii) adhesion, (iii) aggregation, and (iv) biofilm formation. LP- and PK-type compounds synthesized by *B. velezensis* have an effect on pathogenic microbes in the rhizosphere (v) and overall plant growth promotion and defense response of the plants increased (vi).

Antagonism between the two bacterial strains in terms of the swarming assay influences the cooperative behavior of *B. velezensis* SQR9 (Δ*spoA*) + *B. velezensis* FZB42 in the cucumber rhizosphere, which is advantageous for plants, especially for the enhanced production of IAA, acetoin, and LPs. Deletion of the *spoA* gene diminishes the likelihood of antagonism in SQR9, required for sporulation, antimicrobial chemical synthesis, and the production of extracellular matrix [75]. Halotolerant *B. velezensis* XT1, isolated from a saline habitat in Spain, exhibited plant-growth-promoting activities when introduced directly into the soil. Strain XT1 enhanced the aerial fresh weights of tomato, pepper, pumpkin, and cucumber plants by 53, 63.6, 129.2, and 100.8%, respectively. This bacterium is capable of fixing nitrogen, producing siderophores, solubilizing phosphate, and producing many enzymes, including urease and 1-aminocyclopropane-1-carboxylate deaminase, as well as volatile metabolites, such acetoin and 2,-butanediol [76].

Several commercially available *B. velezensis*-based biocontrol agents can shield plants from a wide range of bacterial and fungal diseases. For example, Rhizo Vital® controls *R. solani*, Botrybel controls *B. cinerea*, Serenade® controls *T. aggressivum*, Kodiak™ controls *F. oxyspourum* and *R. solani*, and Taegro® controls *P. infestans* [18]. To combat the fungus *B. cinerea* in tomato leaves and post-harvest fruits, *B. velezensis* 83 is marketed in Mexico as a foliar biofungicide (Fungifree AB™) [72]. *B. velezensis* AK-0 exhibited biocontrol potential against *Colletotrichum gloeosporioides*, which causes bitter rot in apples. The genome of AK-0 exhibits eight potential gene clusters [77]. Pot experiments with *B. velezensis* FJAT-46737 crude LPs (e.g., iturins, fengycins, and surfactins) at a concentration of 1.0 mg/mL exhibited reduced mortality of tomato plants with a control efficacy of 96.2% against bacterial wilt caused by *R. solanacearum* [47].

## 6. Deleterious Nature of *B. velezensis* in Agriculture

In sustainable agriculture, *B. velezensis*-based biocontrol agents are a promising alternative to conventional pesticides; however, the potential negative effects of this microorganism remain poorly understood. *Bacillus* species may swim and swarm (a coordinated behavior in which billions of flagellated bacteria move towards a solid surface) and produce biofilms, all of which are associated with its pathogenicity [78]. In 2008, bacterial rot in onion bulbs was discovered in warehouses in South Korea, and the causal agent was identified as *B. amyloliquefaciens* [79]. However, our laboratory reidentified this pathogen as *B. velezensis* by 16S sequence analysis and comparative research on *B. amyloliquefaciens* [18,79]. In 2017, the pathogen causing soft rot in potatoes was initially identified as *B. amyloliquefaciens* F10-1 based on 16S rRNA sequence analysis; however, molecular characterization of *gyrB* gene clustered *B. amyloliquefaciens* F10-1 as *B. amyloliquefaciens* subsp. *plantarum*, which is actually *B. velezensis* [80]. *B. velezensis* zk1 was the dominant bacterium causing rot in peach fruits [81]. *B. velezensis* zk1 infection lowered the activities of many free-radical-scavenging enzymes, including superoxide dismutase, polyphenol oxidase, catalase, and peroxidase, in the host peach plants, thereby damaging peach chloroplasts, mitochondria, and respiratory chains. These combined actions disrupt the normal physiological metabolism of peach fruits, causing rot [81]. These reports suggest that several *B. velezensis* strains could also act as pathogens in several important agricultural and horticultural crops. Therefore, care must be taken to apply *Bacillus*-based biocontrol agents into the agriculture system and should be investigated broadly prior to application as biofertilizer. However, we did not find any reports of *B. velezensis*-based commercial biocontrol agents that caused disease in agricultural products.

Previously, several other *Bacillus* spp. were also reported to cause diseases in agricultural products. For instance, a *B. cereus* spp. complex consisting of *B. cereus*, *B. thuringiensis*, and *B. pacificus* was responsible for the bacterial leaf spot disease in peach (*Prunus persica* L.), which is cultivated widely in the world [82]. *B. pumilus* was reported to cause fruit rot on muskmelon (*Cucumis melo*) in China [83]. *B. subtilis* is considered to be a universal cell factory for the microbial production of enzymes, chemicals, and antimicrobial metabolites for industry, agriculture, and medicine [84]; however, several *B. subtilis* strains were also reported to cause disease. For example, *B. subtilis* G7, which was isolated from a deep-sea hydrothermal vent, has numerous virulence genes that can kill mice and fish [85]. *B. subtilis* HFBF_B11 isolated from the brain tissue of ducklings caused pathogenicity in animals [86]. *B. cereus* was considered harmless for about 80 years before being accepted as a human pathogen (causing intestinal and extraintestinal diseases) [87].

## 7. Conclusions

The application of biocontrol agents based on *B. velezensis* in agriculture can have several beneficial effects on crop growth and protection. To manage pathogenic phytobacteria, *B. velezensis* can exert strong controlling activities based on its high antimicrobial activities with fast-growing aspects, which can be crucial for its application in sustainable agriculture; however, this bacterium was also identified as the causal agent of several plant diseases. In previous reports, only the beneficial effects of *B. velezensis* were highlighted; however, in this study, we also summarized several harmful reports of this bacterium because this bacterium can be pathogenic to onions [79], potatoes [80], peach fruits [81], etc. Therefore, a more careful approach is required before applying it as a biofertilizer to crop species, cultivars, and breeds in agricultural fields.

**Author Contributions:** M.F.R., B.-S.H. and K.-H.B. collected the data and wrote the manuscript. All authors have read and agreed to the published version of the manuscript.

**Funding:** This research was funded by NNIBR202303105.

**Data Availability Statement:** Not applicable.

**Acknowledgments:** The authors appreciate the research fund provided by the Nakdonggang National Institute of Biological Resources (NNIBR), funded by the Ministry of Environment (MOE) of the Republic of Korea (NNIBR202303105).

**Conflicts of Interest:** The authors declare no conflict of interest.

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
