# Peer review of "Bacillus velezensis: A Beneficial Biocontrol Agent or Facultative Phytopathogen for Sustainable Agriculture"

_agronomy, doi:10.3390/agronomy13030840_

Round 1

Reviewer 1 Report (New Reviewer)

The manuscript is well written and adresses important issues concerning negative roles of bacteria in agriculture. Several minor changes in English are necessary.

Author Response

Reviewer comments -1

The manuscript is well written and addresses important issues concerning negative roles of bacteria in agriculture. Several minor changes in English are necessary.

Response: We appreciate the reviewer's valuable comments. We have edited the manuscript as much as possible to improve this manuscript.  (Please check the English editing certificate), and if needed, we will get another round of English-editing for this manuscript.

Reviewer 2 Report (New Reviewer)

The article is very interesting and contains relevant information on the use of a microorganism in disease management. What is new is that its possible negative effects due to its use are indicated.

A good review of the interaction of the microorganism with the plant and its responses to avoid diseases, as well as physiological, biochemical, and molecular responses, was observed. It is important to send a language editor for review for its best presentation.

Author Response

Reviewer comments-2

The article is very interesting and contains relevant information on the use of a microorganism in disease management. What is new is that its possible negative effects due to its use are indicated.

A good review of the interaction of the microorganism with the plant and its responses to avoid diseases, as well as physiological, biochemical, and molecular responses, was observed. It is important to send a language editor for review for its best presentation.

Response: Response: We appreciate the reviewer's valuable comments. We had edited the manuscript by a professional English editor to check the grammatical points (please check the English editing certificate), and if needed, we will get another round of English-editing for this manuscript.

Reviewer 3 Report (New Reviewer)

In general, the manuscript is conceptually  basic. Aspects of the application of the endophytic organism B. velezenis for crop biocontrol are described far too superficially. In particular, the negative aspects are barely mentioned, so this manuscript should not aim to be a report highlighting the negative aspects of employing B. velezensis in agriculture. The depth of the text could be implemented by introducing comparisons and relationships of biocontrol agents with conventional crop improvement strategies (e.g. biochar, biotechnological applications to the rhizosphere). For this purpose, some example reference articles are given below. Finally, the structure needs to be reviewed from a grammatical point of view. Often, sentences are unclear. Tab 1 is not justified (moreover, why are there highlights in the text?). In conclusion, I think the manuscript is not yet suitable for publication. Significant refinements need to be made.

Arshad, U., Azeem, F., Mustafa, G. et al. Combined Application of Biochar and Biocontrol Agents Enhances Plant Growth and Activates Resistance Against Meloidogyne incognita in Tomato. Gesunde Pflanzen 73, 591–601 (2021). https://doi.org/10.1007/s10343-021-00580-4

Graber, E.R., Meller Harel, Y., Kolton, M. et al. Biochar impact on development and productivity of pepper and tomato grown in fertigated soilless media. Plant Soil 337, 481–496 (2010). https://doi.org/10.1007/s11104-010-0544-6

Tartaglia M., Arena S., Scaloni A., Marra M., Rocco M. Biochar Administration to San Marzano Tomato Plants Cultivated Under Low-Input Farming Increases Growth, Fruit Yield, and Affects Gene Expression. Frontiers in Plant Science (2020)     DOI=10.3389/fpls.2020.01281

Biederman, L.A. and Harpole, W.S. Biochar and its effects on plant productivity and nutrient cycling: a meta-analysis. GCB Bioenergy, (2013)  https://doi.org/10.1111/gcbb.12037

Oluwole OO, Aworunse OS, Aina AI, Oyesola OL, Popoola JO, Oyatomi OA, Abberton MT, Obembe OO. A review of biotechnological approaches towards crop improvement in African yam bean (Sphenostylis stenocarpa Hochst. Ex A. Rich.). Heliyon. 2021 10.1016/j.heliyon.2021.e08481.

Author Response

Reviewer comments -3

In general, the manuscript is conceptually basic. Aspects of the application of the endophytic organism B. velezenis for crop biocontrol are described far too superficially. In particular, the negative aspects are barely mentioned, so this manuscript should not aim to be a report highlighting the negative aspects of employing B. velezensis in agriculture. The depth of the text could be implemented by introducing comparisons and relationships of biocontrol agents with conventional crop improvement strategies (e.g. biochar, biotechnological applications to the rhizosphere). For this purpose, some example reference articles are given below. Finally, the structure needs to be reviewed from a grammatical point of view. Often, sentences are unclear. Tab 1 is not justified (moreover, why are there highlights in the text?). In conclusion, I think the manuscript is not yet suitable for publication. Significant refinements need to be made.

  1. Arshad, U., Azeem, F., Mustafa, G. et al. Combined Application of Biochar and Biocontrol Agents Enhances Plant Growth and Activates Resistance Against Meloidogyne incognita in Tomato. Gesunde Pflanzen 73, 591–601 (2021). https://doi.org/10.1007/s10343-021-00580-4

  1. Graber, E.R., Meller Harel, Y., Kolton, M. et al. Biochar impact on development and productivity of pepper and tomato grown in fertigated soilless media. Plant Soil 337, 481–496 (2010). https://doi.org/10.1007/s11104-010-0544-6

  1. Tartaglia M., Arena S., Scaloni A., Marra M., Rocco M. Biochar Administration to San Marzano Tomato Plants Cultivated Under Low-Input Farming Increases Growth, Fruit Yield, and Affects Gene Expression. Frontiers in Plant Science (2020) DOI=10.3389/fpls.2020.01281

  1. Biederman, L.A. and Harpole, W.S. Biochar and its effects on plant productivity and nutrient cycling: a meta-analysis. GCB Bioenergy, (2013) https://doi.org/10.1111/gcbb.12037

  1. Oluwole OO, Aworunse OS, Aina AI, Oyesola OL, Popoola JO, Oyatomi OA, Abberton MT, Obembe OO. A review of biotechnological approaches towards crop improvement in African yam bean (Sphenostylis stenocarpa Hochst. Ex A. Rich.). Heliyon. 2021 10.1016/j.heliyon.2021.e08481. 

Response: We appreciate the reviewer's valuable comments. We tried to include more information about the negative aspects of B. velezensis in agriculture; however, we did not find enough reports on it. We have concised the paragraph and reworded the structure of the paragraph (please see line no. 368-402). In addition, we have included several other plant diseases caused by B. pumilus, B. cereus in Agriculture to compare with B. velezensis (please see line no. 368-402).

We have checked the articles related to Biochar. As this concept is new to me, I found difficulties to include these articles in this manuscript.

Reviewer 4 Report (New Reviewer)

Line 46-47 also with fungi and other organisms.

Line 50 ecofriendly means what? This is not a scientific concept.

Table 1 - the table is not very readable. It should be reformatted so that the columns are better placed and do not merge with each other.

Table 2 - the title of the first column reads: "B. velezensis strains” and B. amyloliqiefaciens is listed in the column. This is explained in the text (line 93-95), but it should be explained, for example, also under the table.

  The information presented in the manuscript (line 355-365) should be presented in tabular form and extended (indication of manufacturer, country, active bacterial strain, etc.). Bayer's Serenade contains the active strain of Bacillus subtilis, not Bacillus velezensis.

  Line 378-394 - this passage is too far from the subject of the manuscript.

  Line 367-394 should be reworded and shortened.

Author Response

Reviewer comments -4

Authors’ Responses to reviewer Comments

We appreciate the reviewer of this manuscript for providing the critical points to improve it. We have revised the sentences in the manuscript to address the valuable suggestions of the reviewer, and provided the answers as follows:

Line 46-47 also with fungi and other organisms.

Response 1: we have revised the sentence.

Line 50 ecofriendly means what? This is not a scientific concept.

Response 1: We appreciate the reviewer's valuable comments. We have deleted “ecofriendly nature”, and revised the sentence as “For example, in recent years, endophytic microbes are gaining attention to the researchers and scientists due to its (ecofriendly nature) ability to produce metabolites that regulate host plant physiology or having pharmacological significance”. (Blue-colored words were newly added).

Table 1 - the table is not very readable. It should be reformatted so that the columns are better placed and do not merge with each other.

Response 1: We have reformatted the Table 1.  

Table 2 - the title of the first column reads: "B. velezensis strains” and B. amyloliqiefaciens is listed in the column. This is explained in the text (line 93-95), but it should be explained, for example, also under the table.

Response 1:  We have changed the Table 2 and added note as “Several B. amyloliquefaciens strains, such as B. amyloliquefaciens SQR9, B. amyloliquefaciens MEP218 and B. amyloliquefaciens FZB42 were reclassified and reported as B. velezensis”.

 The information presented in the manuscript (line 355-365) should be presented in tabular form and extended (indication of manufacturer, country, active bacterial strain, etc.). Bayer's Serenade contains the active strain of Bacillus subtilis, not Bacillus velezensis.

Response 1: We appreciate the editor’s critical points. We have reported in one of our publications (doi:10.3390/molecules25214973) that several B. subtilis-based commercial biocontrol agents like Serenade® (B. subtilis QST713), Kodiak™ (B. subtilis GB03), Taegro® (B. subtilis var. amyloliquefaciens FZB24) were re-categorized as B. velezensis-based biocontrol agents for agricultural applications.

Line 378-394 - this passage is too far from the subject of the manuscript.

Response 1: We appreciate the reviewer's valuable comments. We have revised the passage (please see line no. 368-402).

Line 367-394 should be reworded and shortened.

Response 1: We appreciate the reviewer's valuable comments. We have revised the paragraph and shortened (please see line no. 368-402).

Round 2

Reviewer 4 Report (New Reviewer)

The manuscript is interesting for the scientific community. I would recommend above mentioned publication to acceptation without any corrections.

This manuscript is a resubmission of an earlier submission. The following is a list of the peer review reports and author responses from that submission.

Round 1

Reviewer 1 Report

I have reviewed the review article entitled ‘Bacillus velezensis, a Friend or a Foe for Sustainable Agriculture. In this study, compiled recent research findings on B. velezensis related to the production of antimicrobial substances, volatile organic compounds, induction of disease resistance, and the effect of this bacterium on plant growth promotion and yield. Although previous research indicates that this important resource is also linked to several diseases in crops, including peach fruits, onions, garlic, and potatoes, the negative aspects of this bacterium in terms of its virulence traits to infect crops have not been summarized before. Owing to its rapid growth and high antibacterial activity, B. velezensis is regarded as an ideal biofertilizer for a variety of sustainable agricultural cultivations. However, because of the inherent pathogenicity of this bacterium to several crops, care must be taken when using it in a novel crop cultivation method. However, the captions in Tables and Figures should be amended.

In addition, English is decent but I suggest a thorough review of the manuscript before accepting it for publication. To further improve the text, I suggest the following changes in the manuscript. Please pay attention on the use of full stop and commas

.

1. Abstract
1). Make the title a simple statement.
2). Give the problem statement in a single line.
3). Give a reason for the selection of the current technique.
4). Quantitative data is also important to support your conclusion. Would you please provide some quantitative data in terms of percentage significant increase or decrease in the abstract?
5). Please provide a conclusive conclusion with is withdrawn through research in a single line.
6). Give future prospective in a single line.
7). As per standard suggestions, please avoid using title words as keywords.

Line 13- 14 Several  biocontrol products based on B. velezensis,  previously identified , please change  to Several B. velezensis-based biocontrol products, previously known as, as B. amyloliquefaciens or B. amyloliquefaciens subsp, 

Line 16; antimicrobial substances change to antimicrobials

Line 19; change to crops including peach, onion, garlic and potato,

Line 22 However, because of change to However, due to

Line 23 method change to technique

2. Introduction
1). Please follow the title and improve the introduction in the following sequence as i.e., Foe for Sustainable Agriculture, problem statement, aims of study and hypothesis.
2). Also, provide a novelty statement at the end. What new things authors have done or correlated in this research compared to old ones?
3. Would you please give a single line about the knowledge gap which your research has covered along with the hypothesis statement?

Line 48- 56 ;  To replace chemical agents in agriculture, many studies have  attempted  to identify  

novel  bacterial  strains  that  can  be  used  as  biocontrol  and/or  biofertilizer  alternatives.   Among many known species, Bacillus and Pseudomonas are the most widely used biological alternatives to chemical agents.  Bacillus-based agro-alternatives are gaining attention owing to their spore-forming ability that can resist harsh environmental conditions for an extended period [6]. In addition,  Bacillus  species  reportedly  have biocontrol potential in  the greenhouse, field, and post-harvest stages of fruit because of their contribution to plant  protection through several mechanisms,  including antibiosis, reduced pathogen colonization in the root via competition, and induction of systemic resistance (ISR) in the host plant  [7].    Authors added in above paragraphs only references please added some recently reference and Among many known species? It’s unclear please explain briefly here.

3. Results and Discussion.
1). Very descriptive. Please give only significant results. Also, give mechanistic discussion. It is not a correct way to discuss results based on other scientists' findings. Please elaborate on specified mechanisms which are regulating and result

Line 80, remove derived  

4. Conclusion
12. dd the targeted beneficiary audience who will get benefits from this review.
Also, give clear-cut recommendations

13 In spite this is review article a lack of recent literature (Recent references (last 3 years), therefore the authors should include the most recent references on this subject

5. References

Standardize references

Reviewer 2 Report

This manuscript is a review documenting mainly the beneficial aspects of Bacillus velezensis in agriculture and also includes very few recent studies in which B. velezensis has been reported as a pathogen to plants. The manuscript includes also some preliminary experimental results that suggest that some B. velezensis isolates cause rotting symptoms in onions and garlic bulbs. However, these preliminary results occupy one full page of the manuscript and 2 of the 5 figures included in the manuscript. It is not valid no include, within a review manuscript, experimental results that have not been revised formally in a specialized journal. Those results have basic deficiencies (see below) in terms of phytopatological protocols to clearly identify the real causal agents of plant diseases. The content of the manuscript mainly describes the beneficial aspects of the use of B. velezensis in agriculture (about 70 references) and only 5 that have reported the potential pathogenicity of some isolates of this bacterium, some of them isolated in extreme environments, which are very different to the conditions prevailing in the field. In addition, the review does not report any case in which registered commercial products containing B. velezensis have caused a disease in any crops; therefore, the title is misleading as it suggests that all B. velezensis could be “foes”.

In the experimental part, the authors did not discuss their results. They, for instance, did not explain and discuss why in one set of onions the isolates inoculated caused rotting and in other set of onions did not. The results are clearly not reproducible. Furthermore, the authors did not report the re-isolation of the strains after causing the disease, a fundamental phytopathological test in order to assure that the inoculated bacterium was the causal agent of the disease. Therefore, it is misleading to say in the Conclusions that “In our review, we highlighted that this bacterium can be pathogenic to garlic and onion bulbs…”

The conclusions are also misleading. For example, the authors say that “…although this bacterium was identified as the causal agent of SEVERAL plant diseases [79,80], only the beneficial effects have been highlighted in previous reports”. References [79,80] report that some isolates of B. velezensis were pathogens of potatoes and onions, not to “SEVERAL plant diseases”. Furthermore, the review of the authors also highlights the BENEFICIAL EFFECTS of B. velezensis in agriculture !

General aspects of bacterial virulence have been used in order to justify that B. velezensis can be a “foe”. For example, in the conclusions the authors say: “…based on recent research, it contains virulence factors in the genome and functions as a plant pathogen for several crops.” This claim is not justified as virulence factor are included in the genomes of a wide number of bacteria, and this not necessarily means that they are pathogens. By the way, the caption of figure 4 (one of the 5 figures included) says “Bacterial virulence factors associated with diseases” and what is included in the figure is a general list of “virulence factors” that are also have been associated to its beneficial effects as biofertilizer or biofungicide ! (enzymes, surfactants, toxins, etc.)  Again, the authors are trying to emphasize that B. velezensis is a “foe” with unjustified reasons.

Reviewer 3 Report

The manuscript "Bacillus velezensis, a Friend or a Foe for Sustainable Agriculture?" by Muhammad Fazle Rabbee, Buyng Su Hwang and Kwang-Hyun Baek, aims to highlight the disadvantages of employing B. velezensis in sustainable agriculture. Generally, the manuscript addresses an important issue. However, the extent of the literature review presented in this research and the conclusions are not in accordance with the defined aim of the manuscript and manuscript lack a clear scientific contribution. The major drawback is that the paper presents the results of the authors that were not previously published and the review articles must not include unpublished material.